# Next-Generation Sequencing of Four Mitochondrial Genomes of *Dolichovespula* (Hymenoptera: Vespidae) with a Phylogenetic Analysis and Divergence Time Estimation of Vespidae

**DOI:** 10.3390/ani12213004

**Published:** 2022-11-01

**Authors:** Hang Wang, Qian Wen, Tongfei Wang, Fanrong Ran, Meng Wang, Xulei Fan, Shujun Wei, Zhonghu Li, Jiangli Tan

**Affiliations:** 1Shaanxi Key Laboratory for Animal Conservation/Key Laboratory of Resource Biology and Biotechnology in Western China, College of Life Sciences, Northwest University, 229 North Taibai Road, Xi’an 710069, China; 2Institute of Plant Protection, Beijing Academy of Agriculture and Forestry Sciences, Beijing 100097, China

**Keywords:** high-throughput sequencing, mtDNA, phylogeny, evolutionary history, insect

## Abstract

**Simple Summary:**

Mitochondria are vital organelles found in most eukaryotes that are involved in energy metabolism. The mitochondrial genome (mtgenome) is the genetic material of mitochondria, which is responsible for the coding and translation of proteins needed by organelles. By studying the mtgenomes of insects, we can better comprehend the phylogenetic relationships and molecular evolution of insects. The mtgenomes of four wasps are described in this paper, along with comparative genomic analyses. There are gene rearrangement events and 37 genes in each of the four mtgenomes. The subfamily Stenogastrinae is the sister group to the remaining Vespidae family, and according to our concluding analysis, the genus *Vespa* is more closely linked to the genus *Vespula* than to the genus *Dolichovespula*. We provide new evidence for the two-origin hypothesis of eusociality in the Vespidae. This work will aid us in future research on species evolution and phylogeny in the family Vespidae.

**Abstract:**

The wasp genus *Dolichovespula* (Hymenoptera: Vespidae: Vespinae) is a eusocial wasp group. Due to the taxonomic and phylogenetic issues with the family Vespidae, more genetic data should be gathered to provide efficient approaches for precise molecular identification. For this work, we used next-generation sequencing (also known as high-throughput sequencing) to sequence the mitochondrial genomes (mtgenomes) of four *Dolichovespula* species, viz. *D. flora*, *D. lama*, *D. saxonica*, and *D. xanthicincta* 16,064 bp, 16,011 bp, 15,682 bp, and 15,941 bp in length, respectively. The mitochondrial genes of the four species are rearranged. The A + T content of each mtgenome is more than 80%, with a control region (A + T-rich region), 13 protein-coding genes (PCGs), 22 tRNA genes, and two rRNA genes. There are 7 to 11 more genes on the majority strands than on the minority strands. Using Bayesian inference and Maximum-Likelihood methodologies as well as data from other species available on GenBank, phylogenetic trees and relationship assessments in the genus *Dolichovespula* and the family Vespidae were generated. The two fossil-based calibration dates were used to estimate the origin of eusociality and the divergence time of clades in the family Vespidae. The divergence times indicate that the latest common ancestor of the family Vespidae appeared around 106 million years ago (Ma). The subfamily Stenogastrinae diverged from other Vespidae at about 99 Ma, the subfamily Eumeninae at around 95 Ma, and the subfamily Polistinae and Vespinae diverged at approximately 42 Ma. The genus *Dolichovespula* is thought to have originated around 25 Ma. The origin and distribution pattern of the genus *Dolichovespula* are briefly discussed.

## 1. Introduction

The long-cheeked yellow jacket genus *Dolichovespula* consists of 19 species distributed in Europe/Asia, North Africa, and North America [1,2,3,4,5]. Archer tentatively divided *Dolichovespula* into seven species groups and discussed their distribution in detail [1,2]. However, China, with one of the most speciose faunae of the world, was not well researched at that time. Ten known species belonging to all seven groups as defined by Archer occur in China. More collection information has been added with the development of our study on Chinese vespids in recent years. The nests and all three castes of *D. flora* and *D. stigma* were discovered, and *D. baileyi* Archer, 1987 is now considered to be synonymous with *D. stigma* Lee, 1986 [4,6]. The additional data, as important evidence, improved group division [5,7]. There are still two high-altitude species, *D. lama* and *D. xanthicincta*, with their nests and male data lacking. More data, especially molecular data, is expected to support further research.

The normal insect mitochondrial genome (mtgenome) measures around 16 kb, is double-stranded and closed-circular, and typically contains 13 protein-coding genes (PCGs), 22 tRNA genes, two rRNA genes, and a control region (A + T-rich region) [8,9]. The mtgenome is a suitable molecular marker for molecular evolution, phylogenetic relationships, and species identification due to its maternal inheritance, short coalescence time, rapid mutation rate, and conservative gene components [10,11,12,13]. Frequent gene rearrangements and a high A + T content are common characteristics of the mtgenomes of Hymenoptera [14,15]. To date, 56 annotated mtgenomes representing 34 species of Vespidae are available in the GenBank. Among them, only one species of genus *Dolichovespula* (*D. panda*) has been annotated [9]. Three additional *Dolichovespula* (i.e., *D. sylvestris*, *D. saxonica*, and *D. media*) mtgenomes are found in GenBank. However, no annotated data are available for these three mtgenomes.

In this research, we sequenced the mtgenomes of *Dolichovespula flora*, *D. lama*, *D. saxonica*, and *D. xanthicincta* via high-throughput sequencing and annotated them. General features of the mtgenomes were described. The phylogenetic trees in the genus *Dolichovespula* and in the family Vespidae were constructed and the divergence times of the family Vespidae were estimated.

## 2. Materials and Methods

### 2.1. Sample Gathering and DNA Isolation

Appendix A provides specifics regarding sample gathering. After identification, each specimen was stored in a sealed box and put in a −20 °C freezer. According to the manufacturer’s instructions, mitochondrial DNA was extracted from the legs of each wasp specimen using the DNeasy tissue kit (Qiagen, Hilden, Germany). The entomological collections of the Beijing Academy of Agriculture and Forestry Sciences housed the voucher DNA. The DNA concentration was quantified using Qubit 3.0 (Invitrogen, Life Technologies, Carlsbad, CA, USA).

### 2.2. Mitochondrial Genome Sequencing and Assembly

The mtgenomic sequences of the four *Dolichovespula* species were obtained by high-throughput sequencing (GenBank: OP250139 *D. flora*, OP250140 *D. lama*, OP250141 *D. saxonica*, and OP250142 *D. xanthicincta*). A string of additional NNNs was inserted to indicate that there are gaps in our submissions and that the genome sequences are incomplete. The libraries were sequenced on Illumina Hiseq 2500 with the strategy of 250 paired-ends by BerryGenomics Company (Beijing, China), and constructed on the Illumina TruSeq@ DNA PCR-Free HT Kit with target insert size 500 bp. Trimmomatic version 0.36 [16] was used to delete adapter sequences and trim low-quality bases. Considering the short reads generated in this research, the presumed mitochondrial targets were ascertained by BLAST searching from the raw reads. A Perl script (FastqExtract.pl accessed on 11 March 2021) was used to obtain the targeted mitochondrial reads which were assembled into contigs under IDBA UD version 1.1.1 and Celera Assembler version 8.3rc2 [17]. Geneious version 8.0 was used to conduct the de novo assembly of the mitochondrial contigs [18].

### 2.3. Mitochondrial Genome Annotation and Analysis

The sequences were annotated online by Mitos Web Server [19] with the parameters Genetic Code = “5 Invertebrate” and Reference = “RefSeq 63 Metazoa”. The mtgenomic maps of the four *Dolichovespula* species were generated by OrganellarGenomeDRAW [20]. The PCGs were determined by finding the open reading frames (ORFs) and then performing BLAST searches in GenBank. The tRNA gene loci and secondary structure were predicted using the tRNAscan-SE Search Server version 1.21 [21]. The rRNA genes and A + T-rich regions were identified by their adjacent genes. The nucleotide composition was automatically displayed in the Geneious software. AT- and GC-skews were computed as follows: AT-skew = (A − T%)/(A + T%) and GC-skew = (G − C%)/(G + C%) [22]. The start/stop codon usage of 13 protein-coding genes was calculated by CodonW (written by John Peden, University of Nottingham, Nottingham, UK). The gene arrangements were compared with the putative ancestral arrangement of the insect mitochondrial genome [23].

### 2.4. Phylogenetic Analysis

Combining the available data published on NCBI, sequences of 13 PCGs and two rRNAs (*rrnL* and *rrnS*) of 38 vespids species were used for phylogenetic analyses with an additional two species of Formicidae as outgroups (Appendix A). Sequences of five mitochondrial genes (*cob*, *cox1*, *cox2*, *rrnL*, and *rrnS*) of 14 *Dolichovespula* species were also used to construct the phylogenetic trees of the genus *Dolichovespula* (Appendix A). Appendix A show detailed information for the species included in this study.

The PCGs of each species were translated into amino acids first, and then aligned using ClustalW with the default parameters implemented in MEGA 7.0 [24]. The two rRNA genes were aligned by MAFFTv7.0 online server (the Q-INS-i strategy is selected in iterative refinement methods because it considers the secondary structure of rRNA) [25]. The ambiguous positions in all alignments were deleted using Gblocks v0.91b [26]. Then, these aligned sequences were concatenated using SequenceMatrix v1.7 [27]. The best partitioning strategy and substitution models for each dataset were determined by PartitionFinder 2.1.1 with the following settings: branch lengths as linked; model election as AICc with the greedy algorithm [28]. Due to the predefined partitions of configuration files, the PCG123 dataset was determined to be 39 data blocks (three codon positions of 13 PCGs were separated), and the PCG123R (PCG123 and rRNAs) dataset was determined to be 41 data blocks (39 data blocks for 13 PCGs and 2 data blocks for *rrnL* and *rrnS*, respectively) in the phylogenetic analysis of Vespidae. The partitioning schemes and models are shown in Appendix A. In the same way, the *Dolichovespula* dataset was determined to be 11 data blocks (9 data blocks for the 3 PCGs and 2 data blocks for *rrnL* and *rrnS*, respectively). The partitioning schemes and models are shown in Appendix A. The PCG123 dataset and PCG123R dataset were used in follow-up phylogenetic analyses.

The phylogenetic trees were generated using the Bayesian inference (BI) method by MrBayes v3.2.2 [29]. Four independent Markov chains were run for 10 million metropolis-coupled generations. Samples were taken every 1000 generations, with the first 25% thrown away as burn-in. Maximum-likelihood (ML) analysis was performed on 1000 repetitions using RAxML based on the fast likelihood method [30]. The CIPRES Science Gateway V3.3 was used to operate RAxML [31].

FigTree v1.4.4 (http://tree.bio.ed.ac.uk/software/figtree/, accessed on 10 May, 2021) and iTOL (https://itol.embl.de/#, accessed on 16 June 2021) were used to edit and beautify phylogenetic trees.

### 2.5. Divergence Time Estimation

The divergence time of Vespidae was estimated with BEAST v1.10.4 [32] based on the available data (PCG123R dataset). An uncorrelated lognormal relaxed clock model was employed. The best partitioning strategy and substitution models were determined using PartitionFinder 2.1.1 in the same way as the MrBayes analyses mentioned above, and considering the evaluation models usable in BEAST. The models of clock and tree except that of substitution were linked among partitions. The tree prior was a Yule process with a random starting tree.

The divergence time of the family Vespidae was estimated based on the two fossil-based calibration dates viz., *Solenopsis richteri* and *Myrmica scabrinodis* as a taxon set being used to calibrate node age prior under a normal prior distribution with a standard deviation of 0.5 and a mean of 93.4 [33], *Abispa ephippium* and *Orancistrocerus aterrimus* as another set being used to calibrate the divergence time with a standard deviation of 0.5 and a mean of 92 [34]. For the test runs with MCMC (Markov Chain Monte Carlo) the length was set to 500 million generations and sampling was undertaken every 10,000 generations.

A Tracer v1.7.2 with 10% burn-in was used to test the convergence of the chains to the stationary distribution. A TreeAnnotator v1.10.4 with 20% burn-in and a posterior probability limit of 50% was used to generate the dated maximum clade credibility tree. The final tree was visualized and edited in FigTree v1.4.4.

## 3. Results

### 3.1. General Features of the Genomes

The four mtgenomes we sequenced are partial genomes mainly because of the A + T-rich region variations and the deletion of a few PCGs sequences. A string of NNNs was inserted representing the gaps in the sequences submitted to GenBank. However, to reflect the relative reality of the mtgenomes, this paper does not consider the inserted NNNs when analyzing the four mitochondrial genome sequences.

The mtgenome of each sequenced species is a double-strand of circular molecular DNA consisting of 37 genes (13 PCGs, 22 tRNA genes, and two rRNA genes) and a partial control region (Figure 1). The majority strands (J-strands) of mtgenomes of *D. flora* and *D. saxonica* each have 23 genes, and the minority strands (N-strands) each contain the remaining 14 genes. The J-strands of the mtgenomes of *D. lama* and *D. xanthicincta* have 22 and 24 genes, respectively, and the N-strands contain the remaining 15 and 13 genes, respectively. The base composition of the mtgenomes is revealed using the AT-skew, GC-skew, and A + T content (Table 1).

The assembled mtgenomes of the four species, viz. *Dolichovespula flora*, *D. lama*, *D. saxonica*, and *D. xanthicincta*, were 16,064 bp, 16,011 bp, 15,682 bp, and 15,941 bp long in turn, and their A + T content covered about 82.78%, 82.85%, 83.05%, and 82.21% of the genome, respectively. Each mtgenome has several intergenic regions and gene overlap regions which may increase the stability of the mitochondrial structures [35].

The mtgenome of *Dolichovespula flora* consists of A = 41.51%, T = 41.27%, G = 5.94%, and C = 11.27% (AT-skew = 0.00, GC-skew = −0.30). The A + T content (82.78%) is significantly higher than the G + C content (17.21%). There were 24 intergenic regions totaling 570 bp in length, each measuring 1–76 bp, and eight gene overlap regions totaling 27 bp in length, each measuring 1–8 bp in length. The largest intergenic region was 76 bp, located between *trnI* and *trnM*. There were four pairs of adjacent genes.

The mtgenome of *Dolichovespula lama* consists of A = 41.80%, T = 41.05%, so that A + T = 82.85%, G = 5.58%, C = 11.57%, then G + C = 17.15% (AT-skew = 0.01, GC-skew = −0.35). There were 25 intergenic regions totaling 1074 bp in length, each measuring 1–224 bp, and seven gene overlap regions totaling 27 bp in length, each measuring 1–7 bp in length. The largest intergenic region was 224 bp, located between *trnN* and *trnF*. There were four pairs of adjacent genes.

The mtgenome of *Dolichovespula saxonica* consists of A = 41.83%, T = 41.23%, so that A + T = 83.05%, G = 5.92%, C = 11.03%, then G + C = 16.95% (AT-skew = 0.01, GC-skew = −0.30). There were 25 intergenic regions totaling 1074 bp in length, each measuring 1–224 bp, and seven gene overlap regions totaling 27 bp in length, each measuring 1–7 bp in length. The largest intergenic region was 286 bp, located between *cox1* and *trnL2*. There were six pairs of adjacent genes.

The mtgenome of *Dolichovespula xanthicincta* consists of A = 41.35%, T = 40.86%, so that A + T = 82.2%, G = 6.10%, C = 11.54%, then G + C = 17.64% (AT-skew = 0.01, GC-skew = −0.31). There were 23 intergenic regions totaling 1114 bp in length, each measuring 2–309 bp, and seven gene overlap regions totaling 27 bp in length, each measuring 1–7 bp in length. The largest intergenic region was 309 bp, located between *nad4l* and *trnT*. The second one was 276 bp, located between *trnM* and *trnI*. There were six pairs of adjacent genes.

### 3.2. Protein-Coding Genes and Codon Usage

Total length of 13 PCGs in *Dolichovespula flora* were 10,831 bp, accounting for 67.42% of the whole mtgenome. The gene *nad2* is only 672 bp long, shorter than in the other vespid species (Appendix A), possibly because we do not have the complete sequence of the *nad2*. Compared with the complete mtgenome of *Dolichovespula panda*, there were estimated gaps (marked as NNNs) about 378 bp long in *nad2* and 57 bp long in *nad6*. The A + T content of all 13 PCGs was 80.31% (Table 1). The PCG with the highest A + T content was *nad2* (86.31%) and the lowest was *cox1* (73.51%) (Table 2). Three PCGs (*nad3*, *nad5*, *nad1*) started from the ATA codon, six genes (*cox1*, *atp6*, *cox3*, *nad4*, *nad6*, *cob*) from ATG, and four genes (*nad2*, *cox2*, *atp8*, *nad4l*) from ATT. The PCGs ended with a typical stop codon TAA, except for *cox1* and *nad1*, which ended with the incomplete termination codon TA.

All 13 PCGs in *Dolichovespula lama* were 11,204 bp long, accounting for 69.98% of the whole mtgenome. Compared with *D. panda*, there were estimated gaps 54 bp long in *nad6*. The A + T content of all 13 PCGs was 81.03% (Table 1). The PCG with the highest A + T content was *nad2* (87.84%) and the lowest was *cox1* (72.74%) (Table 2). Three PCGs (*nad3*, *nad4l*, *nad1*) were initiated from the ATA codon, six genes (*cox1*, *atp6*, *cox3*, *nad4*, *nad6*, *cob*) from ATG, and four genes (*nad2*, *cox2*, *atp8*, *nad5*) from ATT. The PCGs ended with a typical termination codon TAA, except for *cox3*, which ended with the incomplete stop codon TA.

The 13 PCGs in *Dolichovespula saxonica* were 10,528 bp in length, accounting for 67.13% of the whole mtgenome. Possibly because we do not have the complete sequence, length of the *cox1* gene is only 848 bp, which is much shorter than in the other vespid species (Appendix A). Compared with *D. panda*, there were estimated gaps 687 bp long in *cox1* and 51 bp long in *nad6*. The A + T content of all 13 PCGs was 80.70% (Table 1). The PCG with the highest A + T content was *nad2* (88.61%) and the lowest was *cox1* (73.30%) (Table 2). One PCG (*nad1*) started from the ATA codon, seven genes (*nad2*, *cox1*, *atp6*, *cox3*, *nad4*, *nad6*, *cob*) from ATG, and five genes (*cox2*, *atp8*, *nad3*, *nad5*, *nad41*) from ATT. The PCGs ended with a typical stop codon TAA, except that *cox1* and *cox3* ended with the incomplete TA.

The 13 PCGs in *Dolichovespula xanthicincta* were 11,222 bp in length, accounting for 70.40% of the whole mtgenome. Compared with *D. panda*, there were estimated gaps 24 bp long in *nad6*. The A + T content of all 13 PCGs was 79.96% (Table 1). The PCG with the highest A + T content was *nad4* (85.38%) and the lowest was *cox1* (74.14%) (Table 2). Three PCGs (*nad3, nad4l, nad1*) started from the ATA codon, six genes (*cox1*, *cox3*, *atp6*, *nad4*, *nad6*, *cob*) from ATG, and four genes (*nad2*, *cox2*, *atp8*, *nad5*) from ATT. PCG *cox3* ended with the incomplete stop codon TA, *nad1* and *nad4* ended with the termination codon TAG, and the remaining genes ended with a typical termination codon TAA.

The values of codon usage in the mtgenomes of the four *Dolichovespula* species show a remarkable bias toward A and T nucleotides. PCGs *nad2*, *nad6*, and *atp8* have the highest levels of A + T content while the *cox1* has the lowest level (Table 2). Among the 13 PCGs of the four mtgenomes, there were three types of start codon (ATT, ATA, and ATG) and three types of termination codon (TAA, TAG, and TA). The frequencies of Phe, Leu, and Ile were noticeably higher than those of other amino acids in the four sequenced mtgenomes, and the most frequent codons for the three amino acids were TTT (Phe), TTA (Leu), and ATT (Ile). Relative synonymous codon usage (RSCU) in the mitochondrial genomes of the *Dolichovespula* species is shown in Appendix A.

In addition, 38 vespids were selected for phylogenetic tree construction with their start codons and stop codons compared in Figure 2.

### 3.3. tRNA and rRNA Genes

Each mtgenome of the four *Dolichovespula* species contained 22 tRNA genes and two rRNA genes. Among them, 13 tRNA genes were found on the J-strand and the remaining nine were on the N-strand in the mtgenomes of *D. lama*. Fourteen tRNA genes were located on the J-strand and 8 tRNA genes on the N-strand in the mtgenomes of *D. flora*, *D. saxonica*, and *D. xanthicincta*. All tRNAs formed a typical secondary structure (cloverleaf structure) except *trnH* lacked the TΨC loop (Appendix A). The initiation of tRNA of *D. xanthicincta* arrangement as *trnY*-*trnM*-*trnI*-*trnQ* was different from the other three species as *trnY*-*trnI*-*trnM*-*trnQ*.

Two rRNAs of the four *Dolichovespula* species were located on the N-strand. The *rrnS* was located between *trnV* and the A + T-rich region, while the *rrnL* was located upstream at *rrnS*, between *nad1* and *trnV*.

The total length of 22 tRNA genes in *Dolichovespula flora*, *D. lama*, *D. saxonica* and *D. xanthicincta* was 1492 bp, 1497 bp, 1495 bp and 1485 bp, in turn. Each tRNA gene was between 62 bp and 73 bp long (Table 1). Interestingly, the shortest tRNA gene of each species was *trnS1* with a length of 63 bp except for *D. flora*, which had a length of 62 bp. However, the longest gene was not the same (Table 2). In the four species, the tRNA gene with the lowest A + T content was *trnK* (about 76%). The gene with the highest A + T content was *trnV* in *D. lama* (95.52%), *D. saxonica* (92.54%) and *D. xanthicincta* (95.59%), but *trnE* (93.94%) in *D. flora* (Table 2).

The rRNA gene *rrnS* in *D. flora*, *D. lama*, *D. saxonica* and *D. xanthicincta* was 762 bp, 765 bp, 761 bp and 786 bp long, respectively, while *rrnL* was 1367 bp, 1384 bp, 1371 bp and 1361 bp long, respectively. A + T contents of the two rRNA are between 84% and 85% in the four species except *rrnS* in *D. saxonica* (82.00%) and in *D. xanthicincta* (79.77%) (Table 2).

### 3.4. Gene Rearrangements

Further study of insect mitochondrial gene rearrangement events (i.e., remote inversion, gene shuffling, local inversion, and transposition) is helpful in better exploring the phylogenetic relationship among species [15,23,36,37,38].

Three gene rearrangement events occurred in the four *Dolichovespula* species (Appendix A), viz. the translocation of *trnY* to upstream of *trnI* (as in *Vespa mandarinia* and *Dolichovespula panda*), translocation of *trnL1* to the region between *trnS2* and *nad1*(as in *Vespa*, *Polistes*, and *D. panda*), and shuffling of *trnQ* and *trnM* (as in *D. panda*) except *D. xanthicincta* shuffling of *trnQ*, *trnM*, and *trnI*. There were additional events different from each other as follows in *D. flora*: shuffling of *trnE* and *trnS1*, the position and orientation of *trnN* and *trnF* were changed; in *D. lama*: shuffling of *trnE* and *trnN*, local inversion of *trnS1*; in *D. saxonica*: shuffling of *trnE* and *trnS1*, remote inversion of *trnN* and *trnF*; in *D. xanthicincta*: shuffling of *trnE* and *trnS1*, the position and orientation of *trnN* and *trnF* were changed.

In the mtgenome of the *Dolichovespula* species, the order of PCGs and rRNAs was relatively conservative. No rearrangement events occurred in this region. There were conserved gene clusters as *cox1*-*trnL2*-*cox2*-*trnK*-*trnD*, *atp8*-*atp6*-*cox3*-*trnG*-*nad3* and *nad5*-*trnH*-*nad4*-*nad4l*. The rearrangement events mainly occurred in three hot spots as A + T-rich region-*nad2*, *nad2*-*cox1* and *nad3*-*nad5*.

### 3.5. Phylogenetic Relationships

We analyzed the datasets mentioned above (PCG123, PCG123R for Vespidae and five selected genes for *Dolichovespula*) by BI and ML methods and obtained the relevant phylogenetic trees. The phylogenetic trees (Figure 3 and Figure 4) produced by the PCG123R and PCG123 datasets revealed the phylogeny of 38 species of Vespidae. The BI- and ML trees generated by the five genes datasets represented the phylogeny of 14 species of *Dolichovsepula* (Figure 5).

On the level of the subfamilies in Vespidae, both trees strongly supported the sister relationship between Stenogastrinae and the other three common subfamilies of Vespidae in the relationships as Eumeninae + (Polistinae + Vespinae) (Figure 3 and Figure 4). Within the subfamily Vespinae, the results indicated that the relationships among genera were (*Vespa* + *Vespula*) + *Dolichovespula*, which was congruent with other research analysis [9,39,40], but challenges the prevailing hypothesis of yellowjacket (*Vespula* + *Dolichovespula*) monophyly [41,42,43,44].

However, the topologies of phylogenetic trees derived from the two datasets were different within the genus *Dolichovespula* (Figure 3 and Figure 4). The PCG123 dataset supported the relationships (*D. panda* + *D. lama*) + ((*D. saxonica* + *D. xanthicincta*) + *D. flora*), whereas the PCG123R dataset supported the relationships ((*D. panda* + *D. lama*) + (*D. saxonica* + *D. xanthicincta*)) + *D. flora*. The position of *D. flora* was poorly supported in the PCG123 cladogram, with a confidence of ML bootstrap values/Bayesian posterior probabilities 45/0.58, respectively, but strongly supported in the PCG123R, with a confidence of 100/1.

Based on five mitochondrial genes (*cob*, *cox1*, *cox2*, *rrnL* and *rrnS*) of 14 species of *Dolichovespula* and two species of *Polistes* as outgroup, two phylogenetic trees (BI and ML) were constructed (Figure 5). In both trees, the five species i.e., *D. panda*, *D. lama*, *D. saxonica*, *D. xanthicincta* and *D. flora*, clustered at the base (except the outgroup); two sister species *D. maculata* and *D. media* combined with *D. sylvestris* constituted a clade, and the relationships as (*D. arctica* + *D. adulterina*) + *D. omissa* were strongly supported. Compared with the species groups divided by Archer [2,7], the species belonging to *adulterina* group and *lama* group were well clustered. However, the position of *D. flora* was poorly supported in the *maculata* group. The cladogram of the BI tree was different from that of the ML tree on the position of *D. arenaria* with *D. albida* + *D. pacifica*. Considering the data confidentiality level, the relationships indicated by the BI tree were more accurate than those indicated by the ML tree.

### 3.6. Estimated Time of Divergence

The PCG123R dataset on the 38 species of Vespidae was selected to estimate the time of divergence. As shown in the cladogram, the most recent common ancestor of Vespidae diverged around 106 Ma. Sequentially, the subsocial subfamily Stenogastrinae was separated from the other subfamilies of Vespidae at about 99 Ma. The mainly solitary subfamily Eumeninae diverged around 95 Ma. The social subfamily Polistinae divided with the eusocial subfamily Vespidae about 42 Ma and the origin of the genus *Dolichovespula* was estimated at around 25 Ma (Figure 3).

## 4. Discussion

The wasp family Vespidae (Hymenoptera), as model taxa in understanding the evolution from solitary to social habit, consists of more than 5000 species [4]. Based on morphological and behavioral characteristics and then combining them with available molecular sequence data, Carpenter [45,46] supported monophyly of the social subfamilies with phylogenetic relationships as Euparagiinae + (Masarinae + (Eumeninae + (Stenogastrinae + (Polistinae + Vespinae)))). A single origin of eusociality and a view that Vespidae contains six subfamilies has been widely supported, but later Schmitz and Moritz argued that Polistinae + Vespinae had a closer relationship with Eumeninae than Stenogastrinae based on cladistic analyses of molecular data (nuclear 28S rDNA and mitochondrial 16S rDNA) [47]. This suggests that eusociality evolved twice. Hines et al. inferred the same opinion based on four nuclear encoded genes (18S and 28S rDNA, abdominal-A, and RNA polymerase II) [48]. Since Pickett and Carpenter [43] reaffirmed their view of one origin of eusociality based on the largest taxon sample including molecular data, and the largest phenotypic character dataset ever compiled, there are still strongly opposed voices based on transcriptome and other molecular studies [34,49,50,51]. Recently, Piekarski et al. divided the family Vespidae into eight subfamilies based on the Maximum-Likelihood tree constructed from 235 loci selected in 163 Vespidae taxa [51]. Two subfamilies Gayellinae and Zethinae sunk by Carpenter were resurrected by Piekarski et al. [45,46,51]. Interestingly, Piekarski et al. showed the phylogenetic relations as Stenogastrinae ((Masarinae + (Gayellinae + Masarinae)) + (Eumeninae + (Zethinae + (Polistinae + Vespidae)))). In other words, the subsocial subfamily Stenogastrinae appeared first, followed by the emergence of the solitary subfamilies, such as Eumeninae. Finally, the eusocial subfamilies Polistinae and Vespinae were separated from other Vespidae. The results lead to the rejection of the single origin of eusociality hypothesis and strongly support the two-origin hypothesis of eusociality in Vespidae. Obviously, the molecular and phenotypic evidence are still in conflict. Piekarski et al. claim that the molecular evidence supports the dual origin, while the single origin hypothesis is more convincing when all available evidence is considered [52]. For a group of 5000 described species, broader sampling and more new information especially molecular data and analyses are expected to help solve the problem. In this study, we added mt DNA of four *Dolichovespula* species and compared it to that of all other known species. As might have been expected, our results corroborated the dual origin hypothesis.

The eusocial subfamily Vespinae consists of four genera, known as hornets (*Provespa* and *Vespa*) and yellowjackets (*Vespula* and *Dolichovespula*), for which phylogenetic relationships remain controversial [3,53]. Based on morphological and behavioral and/or molecular data, some analyses support that yellowjackets (*Vespula* + *Dolichovespula*) are a clade sister to *Provespa*, placing *Vespa* as a clade sister to the remaining vespine genera [41,44,54]. However, Pickett and Carpenter found *Provespa* + *Vespa* as a clade sister to *Vespula* + *Dolichovespula* [43]. By analyzing the amino acid sequences of antigen 5, Pantera et al. drew a neighbor-joining dendrogram, and found that *Vespula* is closer to *Vespa* than to *Dolichovespula* [55]. Inversely, Lopez-Osorio et al., using transcriptomic (RNA-seq) data, found that *Dolichovespula* is more closely related to *Vespa* than to *Vespula*, therefore challenging the prevailing hypothesis that yellowjacket (*Vespula* + *Dolichovespula*) was a clade [53]. In recent years, some scholars [9,39,40] based on mitochondrial genome research speculated a sister group relationship between *Dolichovespula* and a clade formed by *Vespa* and *Vespula* which supported the conclusion of Pantera et al. [55]. In this study, we confirmed the former conclusion of Pantera et al. [55].

By examining the distribution pattern and phylogenetic research on extant species we might uncover some clues concerning the time of evolution of bios. Hymenoptera was speculated to have started to diversify about 281 Ma by analyzing 3256 protein-coding genes in 173 insect species [49]. The fossil record shows that Vespidae evolved at least by the Early Cretaceous around 120–65 Ma [56]. Perrard et al. estimated that the minimum age of the Eumeninae is approximately 90 Ma [57]. Tan et al. mentioned that the origin of the genus *Zethus* could be estimated to be between 90 and 80 Ma, allowing for the Gondwanic distribution pattern [5]. In this study, the estimated time of origin for Vespidae, Eumeninae and Vespinae agreed with the former research [5,56,57]. This is the first estimate of the time of origin of the genus *Dolichovespula* (approximately 25 Ma) and the largest sample of molecular data for *Dolichovespula* used for phylogenetic analysis.

It is assumed that the diversification center of Vespinae lies in the mountainous regions in the subtropics and warm temperate zone of eastern Asia. The area from the eastern Himalaya to southern China harbors the largest number of hornet species, and fits for the Beringian distribution pattern [58,59,60,61]. Tan et al. supposed that Mt. Qin and Ba are the most speciose areas and contain the central area of Vespinae [3]. The genus *Dolichovespula*, *D. panda* located at the base of the life tree, is found only in China (Sichuan, Shaanxi and Ningxia). *D. lama*, *D. flora* and *D. xanthicincta* are distributed from China to its neighboring countries (i.e., India and Nepal for *D. lama*; Myanmar and Korea for *D. flora*; Bhutan and Myanmar for *D. xanthicincta*). *D. maculata*, *D. arctica*, *D. arenaria* and *D. albida* are mainly distributed in North America (USA and Canada) and the other remaining congeners are widely distributed in Eurasia. In phylogeny analysis, we can conclude that the species in China and its surrounding countries are older than those in other regions. Our findings agreed with the hypothesis that the subfamily Vespinae originated in subtropical, warm eastern Asia and that it had a Beringian distribution pattern.

## 5. Conclusions

This study reports the mtgenomes of four *Dolichovespula* species. Among them, the mtgenomes of *D. flora*, *D. lama*, and *D. xanthicincta* are sequenced for the first time. The annotated mitochondrial genomic sequences of Vespidae in GenBank (including our study) suggest that Stenogastrinae is the oldest subfamily within Vespidae, and is located at the base of the phylogenetic trees, which supports the two-origin hypothesis of eusociality in Vespidae. We concluded that the relationships among genera were (*Vespa* + *Vespula*) + *Dolichovespula* in the subfamily Vespinae. In addition, the results suggest that *Dolichovespula* species in China and its adjacent countries are older than those in other regions and support the hypothesis that Vespinae originated in subtropical, warm eastern Asia and that it had a Beringian distribution pattern.

## Figures and Tables

**Figure 1 animals-12-03004-f001:**
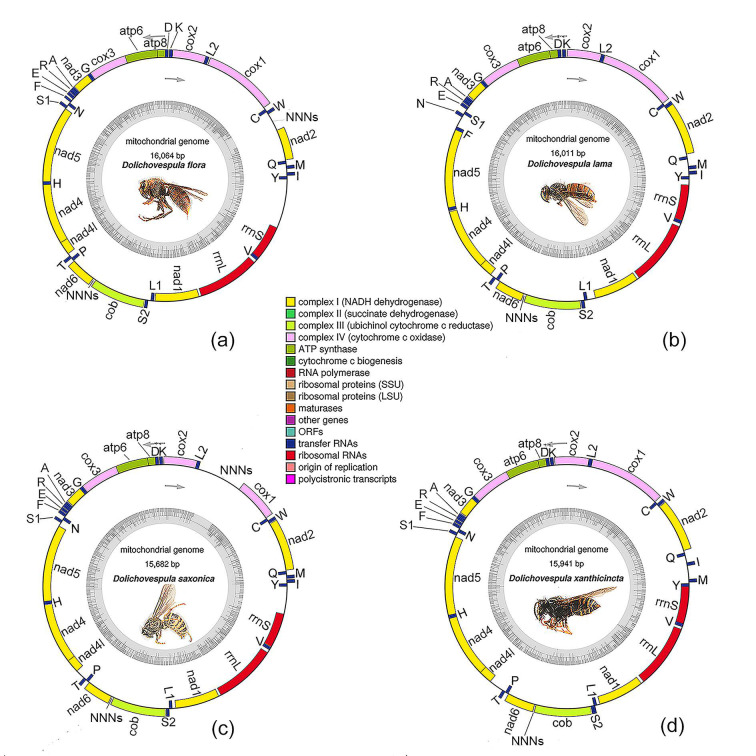
The mitochondrial genomic maps of *Dolichovespula* spp. (**a**) *Dolichovespula flora*; (**b**) *D. lama*; (**c**) *D. saxonica*; (**d**) *D. xanthicincta*. Note: The location of the genes on the direct strand is indicated by colored blocks outside each ring, while the location of the genes on the reverse strand is indicated by colored blocks inside each ring. The NNNs represent the estimated gaps.

**Figure 2 animals-12-03004-f002:**
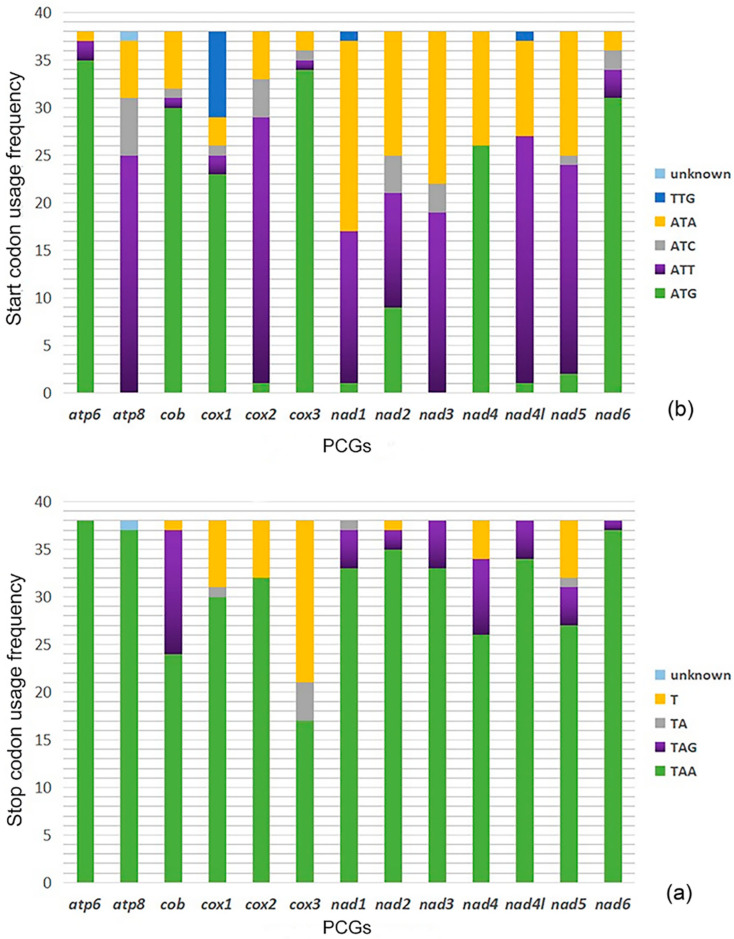
The use of start and stop codons in the mitochondrial genome PCGs of 38 vespids. (**a**) stop codon usage frequency; (**b**) start codon usage frequency. Note: “unknown” stands for the *atp8* gene missing from the *Vespidae* sp. MT 2014.

**Figure 3 animals-12-03004-f003:**
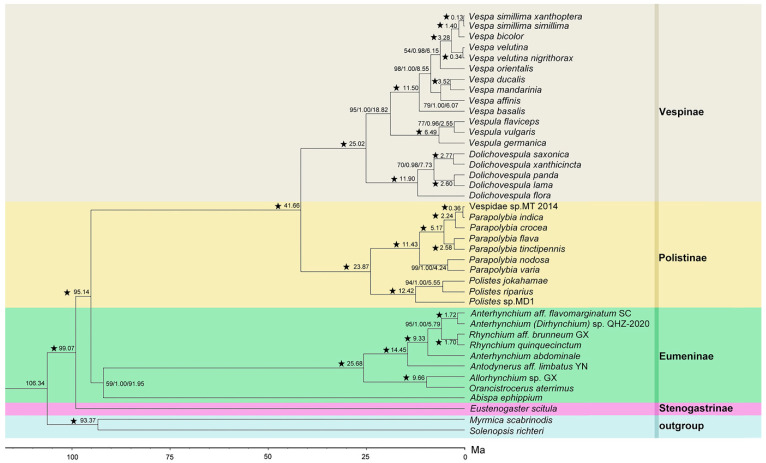
Phylogenetic tree of Vespidae based on the PCG123R dataset by Bayesian inference (BI) and Maximum-Likelihood (ML) methods with evolutionary time scale using beast analysis. Values above the nodes represent divergence time/Bayesian posterior probabilities/ML bootstrap values, respectively. “★” indicates Bayesian posterior probabilities = 1.00 and ML bootstrap = 100. A time scale is shown at the bottom. Ma = one million years ago.

**Figure 4 animals-12-03004-f004:**
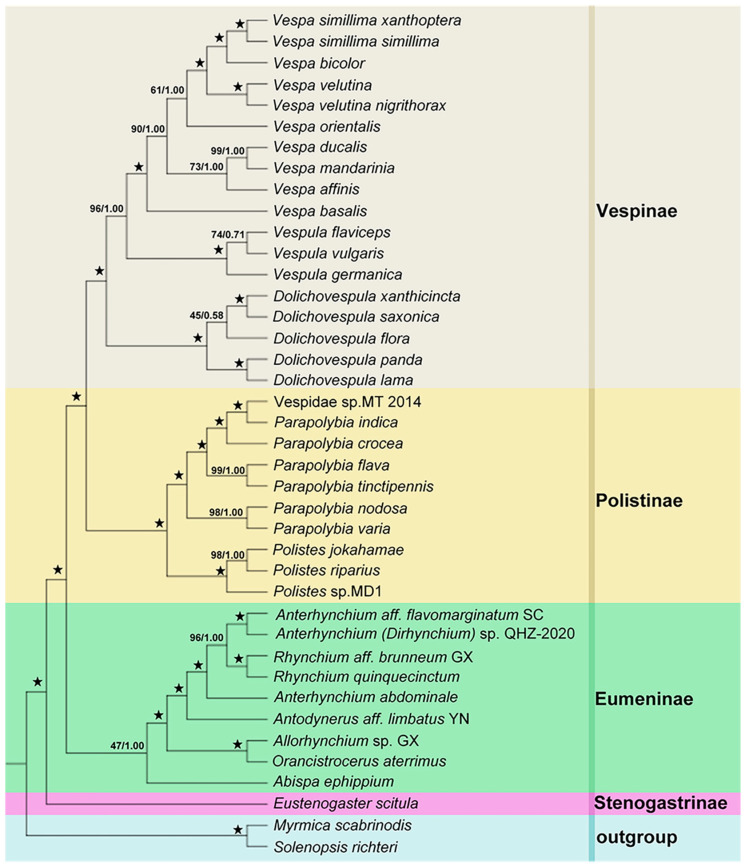
Phylogenetic tree of Vespidae based on the PCG123 dataset by BI and ML methods. Values above the nodes represent Bayesian posterior probabilities/ML bootstrap values. “★” indicates Bayesian posterior probabilities = 1.00 and ML bootstrap = 100.

**Figure 5 animals-12-03004-f005:**
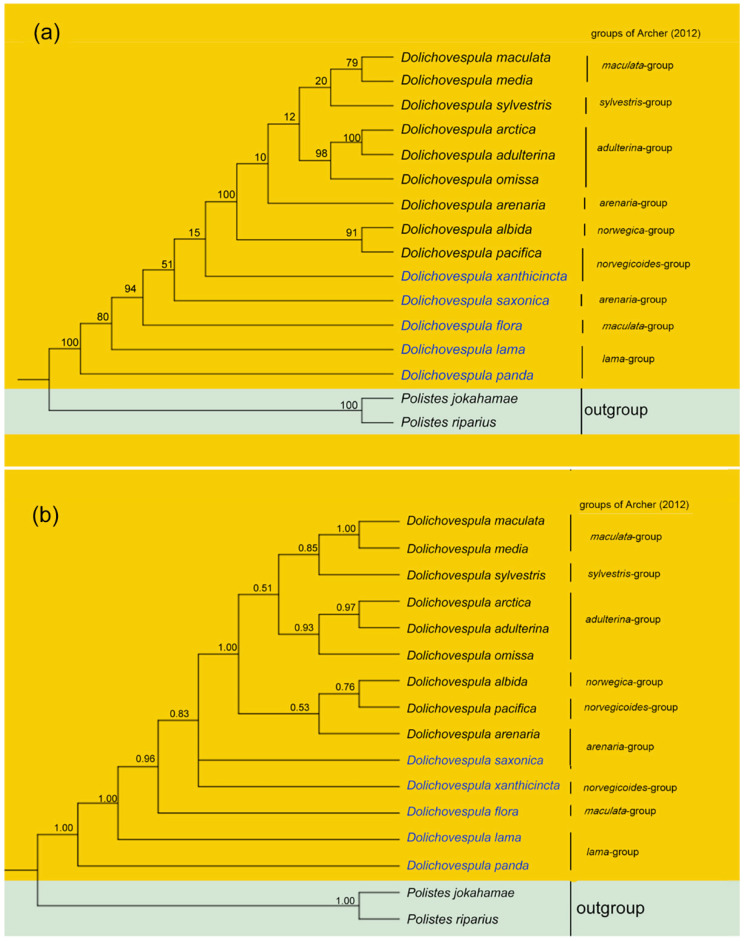
Phylogenetic trees based on five mitochondrial genes (*cob*, *cox1*, *cox2*, *rrnL* and *rrnS*) of 14 species of *Dolichovespula* with two *Polistes* sp. as the outgroup. The species groups of Archer (2012) were noted. (**a**) under Maximum Likelihood (ML); (**b**) under Bayesian inference (BI). Values above the nodes represent bootstrap values (**a**) or Bayesian posterior probabilities (**b**).

**Table 1 animals-12-03004-t001:** General features of the genomes of the four *Dolichovespula* species.

Species	*D. flora*	*D. lama*	*D. saxonica*	*D. xanthicincta*
Whole length (bp)	16,064	16,011	15,682	15,941
Protein-coding genes	13	13	13	13
tRNA genes	22	22	22	22
rRNA genes	2	2	2	2
Majority strand	23	22	23	24
Minority strand	14	15	14	13
Whole (A + T)%	82.78	82.85	83.05	82.21
A%	41.51	41.8	41.83	41.35
T%	41.27	41.05	41.23	40.86
G%	5.94	5.58	5.92	6.10
C%	11.27	11.57	11.03	11.54
AT-skew	0.00	0.01	0.01	0.01
GC-skew	−0.30	−0.35	−0.30	−0.31
PCGs length (bp)	10,831	11,204	10,528	11,222
PCGs (A + T)%	80.31	81.03	80.70	79.96
tRNA length (bp)	1492	1497	1495	1485
tRNA (A + T)%	86.48	86.32	85.03	85.62
rRNA length (bp)	2129	2149	2132	2147
rRNA (A + T)%	84.40	84.69	83.35	83.28
A + T-rich region length(bp)	1069	114	593	309
A + T-rich region (A + T)%	93.08	91.23	92.41	92.88
Gene overlap region	8	7	7	7
Range (bp)	1–8	1–7	1–7	1–7
Size (bp)	27	27	28	27
intergenic region	24	25	23	23
Range (bp)	570	1074	962	1114
Size (bp)	1–766	1–224	3–286	2–309
Adjacent genes	4	4	6	6

**Table 2 animals-12-03004-t002:** Nucleotide composition of 13 PCGs and 2rRNAs of four *Dolichovespula* species.

** *Dolichovespula flora* **						** *Dolichovespula lama* **					
Gene	T%	C%	A%	G%	A + T%	Gene	T%	C%	A%	G%	A + T%
*nad2*	46.13	11.16	40.18	2.53	86.31	*nad2*	47.96	9.50	39.89	2.66	87.84
*cox1*	39.73	14.14	33.78	12.35	73.51	*cox1*	38.65	14.72	34.09	12.54	72.74
*cox2*	39.43	13.24	38.84	8.48	78.27	*cox2*	41.92	12.57	38.52	6.99	80.44
*atp8*	41.11	10.56	42.78	5.56	83.89	*atp8*	41.08	9.73	43.24	5.95	84.32
*atp6*	44.74	12.46	36.94	5.86	81.68	*atp6*	44.28	12.88	36.76	6.08	81.04
*cox3*	42.26	16.67	33.93	7.14	76.19	*cox3*	42.77	13.20	35.28	8.76	78.05
*nad3*	44.75	16.30	32.87	6.08	77.62	*nad3*	43.13	14.01	36.54	6.32	79.67
*nad5*	50.74	6.55	30.80	11.90	81.55	*nad5*	50.14	6.27	30.77	12.82	80.91
*nad4*	49.85	5.95	32.14	12.05	81.99	*nad4*	49.38	5.32	32.76	12.54	82.15
*nad4l*	49.52	2.22	36.19	12.06	85.71	*nad4l*	52.50	1.88	34.38	11.25	86.88
*nad6*	50.17	10.20	35.79	3.85	85.95	*nad6*	47.10	9.84	40.25	2.81	87.35
*cob*	42.26	13.24	33.18	11.31	75.45	*cob*	42.92	13.49	34.28	9.31	77.21
*nad1*	45.54	5.80	35.12	13.54	80.65	*nad1*	49.02	5.06	32.92	13.00	81.94
*rrnL*	42.28	10.82	42.28	4.60	84.56	*rrnL*	43.21	10.55	41.76	4.48	84.97
*rrnS*	44.20	11.08	39.90	4.82	84.09	*rrnS*	42.75	10.98	41.44	4.84	84.18
** *Dolichovespula saxonica* **						** *Dolichovespula xanthicincta* **					
Gene	T%	C%	A%	G%	A + T%	Gene	T%	C%	A%	G%	A + T%
*nad2*	48.31	8.57	40.30	2.82	88.61	*nad2*	49.14	10.38	36.48	4.00	85.62
*cox1*	39.42	14.17	33.88	12.52	73.30	*cox1*	38.86	15.05	32.29	13.81	71.14
*cox2*	39.91	13.79	38.03	8.27	77.94	*cox2*	40.06	13.74	37.72	8.48	77.78
*atp8*	43.21	10.49	43.21	3.09	86.42	*atp8*	45.67	10.10	36.06	8.17	81.73
*atp6*	45.32	12.33	36.26	6.09	81.58	*atp6*	44.29	13.21	36.64	5.86	80.93
*cox3*	44.29	12.69	33.76	9.26	78.05	*cox3*	41.78	14.24	35.50	8.48	77.28
*nad3*	45.33	14.01	34.07	6.59	79.40	*nad3*	46.26	13.85	33.80	6.09	80.06
*nad5*	48.59	6.78	32.11	12.52	80.70	*nad5*	49.05	6.57	31.52	12.86	80.57
*nad4*	48.78	5.74	32.49	12.99	81.26	*nad4*	47.62	5.90	33.71	12.76	81.33
*nad4l*	50.48	2.56	34.82	12.14	85.30	*nad4l*	50.48	2.89	35.05	11.58	85.53
*nad6*	46.55	10.00	38.45	5.00	85.00	*nad6*	46.26	9.15	41.26	3.33	87.52
*cob*	42.28	12.90	35.40	9.42	77.68	*cob*	42.76	12.38	34.76	10.10	77.52
*nad1*	46.96	5.26	34.67	13.11	81.63	*nad1*	47.90	5.13	33.95	13.03	81.85
*rrnL*	43.11	10.58	40.99	5.32	84.10	*rrnL*	43.72	10.07	41.59	4.63	85.30
*rrnS*	41.66	12.22	40.34	5.78	82.00	*rrnS*	41.35	11.83	38.42	5.34	79.77

## Data Availability

The data underlying this article are available in the GenBank Nucleotide Database at https://www.ncbi.nlm.nih.gov/genbank/ (accessed on 16 December 2021), and can be accessed with accession numbers OP250139, OP250140, OP250141, OP250142.

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
