# Peer review of "Next-Generation Sequencing of Four Mitochondrial Genomes of Dolichovespula (Hymenoptera: Vespidae) with a Phylogenetic Analysis and Divergence Time Estimation of Vespidae"

_animals, 2022, doi:10.3390/ani12213004_

Round 1

Reviewer 1 Report

I have carefully read the manuscript and the data is important and the discussion is interesting. I have some concerns that I highlight in the pdf file. 

Hope they are helpful. 

Best.

Author Response

Dear reviewer:

Thank you very much for your comments.

We have taken note of every issue you brought up and fixed it.

We invited an expert to polish the manuscript to improve the grammar and clarity as per your suggestion.

We made so many changes to the manuscript that it was almost like rewriting the entire article. As a result, the revised manuscript did not use the “Track Changes” function because it was a bit redundant and made the manuscript look too confusing. We hope to have your understanding.

We prvide a point-by-point response to the your comments in the attachment.

Sincerely,

Hang Wang
Xi'an, Shaanxi Province, China

Reviewer 2 Report

The authors sequenced the mitochondrial genomes of four Dolichovespula species by the next-generation sequencing technique. However, I think these four mt genomes are not complete. The authors also inserted a string of nnns to represent the gaps in their submissions. But I cannot access these submissions in NCBI. I strongly doubt they missed some sequences in the control region. For example, the authors annotate CR between ND4L and trnT in D. xanthicincta. As CR are located between rrnS and trnY in all other sequenced Dolichovespula species, I think the authors did not get sequences between rrnS and trnY in D. xanthicincta. They should carefully check the submissons, or described the four mt genomes as partial gennomes in the revised MS. 

Author Response

Dear reviewer:

Thank you for your reply and comments.

We have taken note of every issue you brought up and fixed it.

We invited an expert to polish the manuscript to improve the grammar and clarity as per your suggestion.

We made so many changes to the manuscript that it was almost like rewriting the entire article. As a result, the revised manuscript did not use the “Track Changes” function because it was a bit redundant and made the manuscript look too confusing.

We have provided a point-by-point response to your comment in the attachment.

Sincerely,

Hang Wang
Xi'an, Shaanxi Province, China

Reviewer 3 Report

The subject manuscript is a worthy contribution to the insect mitogenomic literature, and makes valuable contributions that will be useful for future studies. The manuscript is in need to grammatical correction throughout, and at times the awkward grammatical constructions interfered with meaning, making them difficult to correct. I started to make grammar suggestions (most of lines 1 – 12 below), but I stopped after the abstract, as I think the paper needs a thorough grammatical review by somebody familiar with the actual study.

1.       Line 4 (title): change “estimating” to “estimation.”

2.       Lines 16-18: Correct as follows:  “The subfamily Stenogastrinae is the sister group of all remaining the family Vespidae, and the genus Vespa shows up as is more closely related to the genus Vespula than to the genus Dolichovespula in our conclusions analysis.”

3.      Lines 18-19: Correct phrasing as: “We provide new support for the two-origin hypothesis of eusociality in Vespidae.”

4.      Line 22: Use of the term “highly developed” seems to be redundant here. Eusociality the most highly developed form of sociality.

5.      Line 25: change “by the” to “using”

6.      Line 28: what is “the ancestral insect”? Do you mean the hypothetical common ancestor of Vespidae, or are you referring to an actual outgroup reference genome? [This is eventually explained in the results; it’s the Drosophila yakuba reference genome. The wording here is still awkward].

7.      Line 28: Change “were” to “are.” Generally, it sounds best to use present tense when referring to actions you perform in the manuscript (like describe results), and past tense when you are describing actual research methods that you used. For example:

a.      “In this manuscript, we describe the results of a Bayesian phylogenetic analysis”

b.      “We used Bayesian phylogenetic methods to reconstruct the phylogeny”

8.      Line 28: what is the difference between “construction” and “reconstruction” of a phylogeny in this context?

9.      Lines 31-32: It is not clear what is meant by “Two fossil-calibrated divergence data…” Do you mean two fossils-based calibration dates were used?

10.  Line 33: change “Divergence time of clades shows…” to “Divergence times indicate…”

11.  Line 35: change “were” to “are”

12.  Line 36: Most of the words in the keywords are also in the title. My understanding is that (usually) key words are used for indexing to improve search results in literature databases. Because titles are also indexed, overlap between keywords and the title are a wasted use of the keywords section.

13.  Line 59: As written, this is a little misleading. Perhaps it is true that the Dolichovespula panda mitochondrial genome has been “reported” (reference 9), but there are three additional Dolichovespula mitochondrial genomes in GenBank: D. sylvestris, D. saxonica, and D. media. Therefore, the D. saxonica mitogenome presented in this study is not actually the first, as implied in lines 60-61.

14.  Line 69: change “refrigerator” to “feezer”

15.  Awkward grammatical constructions appear throughout. For example, it is not clear what is meant by, “The relationships as (D.arctica + D. adulterina) + D. omissa were strongly supported either” (lines 378-379).

16.  Lines 400-402: This is based on an older classification. The work of Piekarski et al. (2018, citation number 51) resurrects the subfamily names sunk by Carpenter (1981, reference number 45). So, in the newer classification, there are eight subfamilies of Vespidae: Stenogastrinae, Gayellinae, Euparagiinae, Masarinae, Eumeninae, Zethinae, Vespinae, and Polistinae. The authors need not necessarily agree with this newer classification, but they should discuss it when talking about the number of extant subfamilies. Also, are there any issues associated with taxon sampling that may affect the results or conclusions? 

Author Response

Dear reviewer:

Thank you very much for your reply and comments.

We have taken note of every issue you brought up and fixed it.

We invited an expert to polish the manuscript to improve the grammar and clarity as per your suggestion.

We made so many changes to the manuscript that it was almost like rewriting the entire article. As a result, the revised manuscript did not use the “Track Changes” function because it was a bit redundant and made the manuscript look too confusing. 

We have provided a point-by-point response to your comments in the attachment.

Sincerely,

Hang Wang
Xi'an, Shaanxi Province, China

Round 2

Reviewer 2 Report

1.      The authors should check the submission of Dolichovespula flora as “UNVERIFIED” was showed in NCBI website. They should clearly describe the mtgenomes are partial in the beginning of Results rather than in the end of discussion. Moreover, the Fig 1 and Table S8 should reflect where are the NNNs. I checked the four submissions in GenBank and I also found there are 687 estimated gaps (NNNs) between cox1 and trnL2 in Dolichovespula saxonica, but these NNNs gaps were not described in MS and the Supplementary Tables.

2.      There is no need to describe the A + T-rich regions in a separate paragraph as the regions were not completely sequenced in this study.

3.      “mtgenomes” rather than “mtGenomes”.

4.      It is awkward to describe each of mtgenomes in Line 244-278. I can get this information in Tables. The authors should compare and summarize these data.

Author Response

Dear reviewer:

Thank you for your reply and comments.

We have taken note of every issue you brought up and fixed it.

The problem with the submission of Dolichovespula flora as “UNVERIFIED” has also been solved, and the updated sequence will be released in a period of time.

Please see the attachment for more details.

Sincerely,

Hang Wang
Xi'an, Shaanxi Province, China
